# Identifying Genes Associated with Female Flower Development of *Phellodendron amurense* Rupr. Using a Transcriptomics Approach

**DOI:** 10.3390/genes14030661

**Published:** 2023-03-06

**Authors:** Lihong He, Yongfang Fan, Zhao Zhang, Xueping Wei, Jing Yu

**Affiliations:** Institute of Medicinal Plant Development, Chinese Academy of Medical Sciences, Peking Union Medical College, Beijing 100193, China

**Keywords:** DNA methylation, *Phellodendron amurense*, female flower development, transcriptome, epigenetics, plant sex differentiation, sex-related genes

## Abstract

*Phellodendron amurense* Rupr., a species of Rutaceae, is a nationally protected and valuable medicinal plant. It is generally considered to be dioecious. With the discovery of monoecious *P. amurense*, the phenomenon that its sex development is regulated by epigenetics has been revealed, but the way epigenetics affects the sex differentiation of *P. amurense* is still unclear. In this study, we investigated the effect of DNA methylation on the sexual development of *P. amurense*. The young inflorescences of male plants were treated with the demethylation agent 5-azaC, and the induced female flowers were obtained. The induced female flowers’ morphological functions and transcriptome levels were close to those of normally developed plants. Genes associated with the development of female flowers were studied by comparing the differences in transcriptome levels between the male and female flowers. Referring to sex-related genes reported in other plants, 188 candidate genes related to the development of female flowers were obtained, including sex-regulating genes, genes related to the formation and development of sexual organs, genes related to biochemical pathways, and hormone-related genes. *RPP0W*, *PAL3*, *MCM2*, *MCM6*, *SUP*, *PIN1*, *AINTEGUMENTA*, *AINTEGUMENTA-LIKE6*, *AGL11*, *SEUSS*, *SHI-RELATED SEQUENCE 5*, and *ESR2* were preliminarily considered the key genes for female flower development. This study has demonstrated that epigenetics was involved in the sex regulation of *P. amurense*, with DNA methylation as one of its regulatory modes. Moreover, some candidate genes related to the sexual differentiation of *P. amurense* were obtained with analysis. These results are of great significance for further exploring the mechanism of sex differentiation of *P. amurense* and studying of sex differentiation of plants.

## 1. Introduction

As the sex organ of plants, floral organs play an important role in plant reproduction. Flower development begins with flower bud differentiation. During the development of flower buds, the meristems differentiate into different flower organ primordia first and then develop into corresponding flower organs. Flower buds can be divided into bisexual flowers and unisexual flowers according to the development of pistils and stamens, and unisexual flowers can be divided into female and male flowers according to the abortion of stamens or pistils. Genetic factors, environmental conditions, hormones, and other factors can regulate the sex differentiation in flowers. Among them, epigenetics has been widely considered to play a crucial role in plant sexual differentiation. DNA methylation is one of the most intensively studied modes of epigenetic regulation [1,2].

DNA methylation is a universal regulatory mode in plants and animals. DNA methylation is the process of transferring methyl groups to a DNA nucleotide catalyzed by DNA methyltransferase with S-adenosylmethionine as methyl donor, which can always influence the expression of target genes [3]. DNA methylation can regulate plant sex by modifying sex-determining genes. As the sex switch in poplar, the *ARR17* gene is silenced by male-specific DNA methylation [4]; the transition from male to female flowers in gynoecious melon results from DNA methylation in the promoter of *CmWIP1* that can contribute to the development of male flowers through carpel abortion [5]; and the male flowers of *Diospyros kaki* are closely connected with the low expression of the female-determining gene *MeGI* because of DNA methylation [6,7]. In addition, DNA methylation can regulate plant sex by modifying sex-related genes in response to external cues. DNA methylation of *CpHUA1* in papaya in response to temperature-related stress can lead to sex reversal from male to hermaphrodite [8]. The sex of cucumber is unstable and easily influenced by temperature and photoperiod. DNA methylation plays a key role in regulating cucumber sex by affecting sex-related genes in response to external conditions [9,10]. With the development of research technology, an increasing number of plants have been reported to regulate their sex with epigenetics.

*P. amurense* Rupr., a lofty deciduous arbor in the Rutaceae, is a valuable medicinal tree species with high economic value [11]. It is recorded that *P. amurense* is a dioecious plant with terminal inflorescences. The stamens or pistils degenerate during the development of unisexual flowers [12]. However, monoecious *P. amurense* was first discovered by Fan et al. [13]. The top of monoecious *P. amurense* died when it was young, and the lateral buds grew and developed into branches of different sexes, which suggested that the sex formation of *P. amurense* occurred after the formation of branches, and there was different sex expression in the same genetic background. The sex expression of *P. amurense* is closely related to epigenetic regulation. DNA methylation is a common epigenetic regulatory mode that has been extensively studied. Therefore, we aimed to explore the sexual differentiation in *P. amurense* through demethylation in this study.

The common demethylation reagent 5-azacytidine (5-azaC) can specifically inhibit DNA methyltransferase from preventing DNA methylation [14,15]. Many studies have shown that 5-azaC can significantly reduce the methylation level in plants and affect their growth and development [16,17,18]. This method is also gradually applied in the study of plant sexual differentiation. Research on *Melandrium album* has shown that treatment with 5-azaC could induce a sex change to andromonoecy in about 21% of male plants, while no apparent phenotypic effect was observed in females [19]. In this study, we treated male flowers of *P. amurense* with 5-azaC and found that some of these male flowers had undergone sex transition, forming 5-azaC-induced female flowers.

In this study, the generation of 5-azaC-induced female flowers suggests that the sex of *P. amurense* is regulated by epigenetics. To explore genes related to the development of female flowers, we have compared the transcriptome data of female flowers and induced female flowers with male flowers, in combination with the sex-related genes reported in the literature. The results of this study can lay a foundation for further exploring the sexual differentiation mechanism of *P. amurense* and provide important information for the study of plant sexual differentiation.

## 2. Materials and Methods

### 2.1. Plant Materials and 5-azaC Treatment

The flower buds of 11 male *P. amurense* individuals preserved in Beijing Medical Botanical Garden were treated with 1 mM 5-azaC in sterile distilled water (floral spray of about 0.3 mL 5-azaC to every inflorescence), which continued during the dormancy period (16 November 2021–10 January 2022) and the growth period (8 March 2022–20 April 2022) once every 2 days. Twenty-eight and twenty-two treatments were performed in the dormant period and the growing period, respectively.

### 2.2. Samples Collection

When the sex of flowers could be accurately distinguished in appearance (diameter of flower buds of about 5 mm), induced female flowers (F_i_) and male flowers without sex conversion after treatment (M_5-azaC_) were collected, and male flowers without 5-azaC treatment on the same plant were collected as controls (male flowers for CK; M_ck_). Female flowers (F) and male flowers (M) of nontreated plants were collected simultaneously. Each sample had three biological repeats. The samples above were immediately frozen in liquid nitrogen and stored at −80 °C.

### 2.3. RNA Isolation, Illumina Sequencing, Transcriptome Assembly, and Annotation

Total RNAs were extracted with TRIzol reagent (Invitrogen, Carlsbad, CA, USA) following the manufacturer’s protocol. The quality and quantity of RNA samples were assessed using the RNA Nano 6000 Assay Kit of the Bioanalyzer 2100 system (Agilent Technologies, Santa Clara, CA, USA).

Total RNA was used as input material for the RNA sample preparations. Poly (A) mRNA was purified from total RNA using poly-T oligo-attached magnetic beads. Then, a cDNA library was constructed. First-strand cDNA was synthesized using random hexamer primer and M-MuLV Reverse Transcriptase, then using RNaseH to degrade the RNA. Second-strand cDNA synthesis was subsequently performed using DNA Polymerase I and dNTPs. After passing the library quality inspection, the library preparations were sequenced on an Illumina Novaseq 6000 platform (Illumina, San Diego, CA, USA), and 150 bp paired-end reads were generated. Clean data (clean reads) with high quality were obtained by removing reads containing adapters, reads containing N base, and low-quality reads from raw data. Transcriptome assembly was accomplished using Trinity v2.4.0 [20]. The longest non-redundant unigenes were acquired by removing sequence splicing and redundancy using Corset v4.6 [21]. The assembly quality was assessed with BUSCO [22].

For further annotation of unigenes, all the assembly unigenes were first searched against the NCBI non-redundant protein sequences (Nr) using diamond v0.8.22 (e-value < e^−5^) [23]. Then, Blast2GO v2.5 [24] was used to obtain the Gene Ontology (GO) annotation of unigenes based on the Nr annotation (e-value < e^−6^). Pathway assignments were carried out according to the Kyoto Encyclopedia of Genes and Genomes (KEGG) pathway database using the KEGG Automatic Annotation Server (e-value < e^−10^) [25].

### 2.4. Acquisition of Sex-Related Genes in P. amurense

It has been shown that the underlying mechanisms controlling flower development are largely conserved in distantly related dicotyledonous plant species [26]. Therefore, genomic resources generated from other plants could be used to identify the potential genes involved in the sexual differentiation and flower development of *P. amurense*. A literature survey was undertaken to list 1281 genes involved in sex differentiation and flower development in other plants, involving 189 papers from more than 30 journals (the latest published in 2022) [4,8,27,28,29,30,31,32,33,34,35,36,37,38,39,40,41,42,43,44,45,46,47,48,49,50,51,52,53,54,55,56,57,58,59,60,61,62,63,64,65,66,67,68,69,70,71,72,73,74,75,76,77,78,79,80,81,82,83,84,85,86,87,88,89,90,91,92,93,94,95,96,97,98,99,100,101,102,103,104,105,106,107,108,109,110,111,112,113,114,115,116,117,118,119,120,121,122,123,124,125,126,127,128,129,130,131,132,133,134,135,136,137,138,139,140,141,142,143,144,145,146,147,148,149,150,151,152,153,154,155,156,157,158,159,160,161,162,163,164,165,166,167,168,169,170,171,172,173,174,175,176,177,178,179,180,181,182,183,184,185,186,187,188,189,190,191,192,193,194,195,196,197,198,199,200,201,202,203,204,205,206,207,208,209,210,211,212,213,214,215]. These can be divided into the following four categories according to function: sex regulatory genes (Group A), including sex-determining genes, genes involved in sex regulatory mechanisms, and genes located in sex-determining regions (SDR); genes related to floral initiation and development (Group B), such as genes related to floral primordium formation and floral organ development; genes related to biochemical metabolic pathways (Group C); and genes related to plant hormones (Group D) (Appendix A).

Nucleotide sequences of the above sex-related genes reported in the literature were downloaded from the NCBI Genbank database in FASTA format. A series of BLASTN analyses using e-value < e^−10^ as a threshold identified broadly conserved sequences of potential sex-related genes from the *P. amurense* transcriptome [216]. The sequences acquired with BLASTN were input to the Conserved Domains platform of NCBI to ensure the homologous sequences shared the same domains.

### 2.5. Differential Expression Analysis

RSEM was used for transcript abundance estimation [217]. Pre-processed RNA-Seq paired reads for each flower sex type were mapped to the final assembled transcriptome using Trinity. All read counts were calculated using the fragments per kilobase of transcript per million fragments mapped (FPKM) method.

Differential expression analysis was conducted between female and male flowers in nontreated trees. To obtain information on sex-related genes of the female and male *P. amurense*, differentially expressed genes (DEGs) between female and male flowers were identified by comparing F with M using DESeq2 [218] based on criteria set as a |log2 fold change| ≥ 1 and a false discovery rate (FDR) ≤ 0.05. Further, we mainly aimed at the sex-related genes identified in *P. amurense* transcriptome using blasting to screen genes in a small scope. Therefore, we conducted a differential expression analysis of sex-related genes between F and M using FDR ≤ 0.05 as the criterion.

An analysis of genes related to female flower development was carried out. First, differential expression analysis of sex-related genes between F_i_ and M_ck_ was conducted to obtain sex-related genes that changed during sex conversion after treating male flowers with 5-azaC. The FDR ≤ 0.05 was used as a threshold. Next, considering that some genes unrelated to sex might also be changed with 5-azaC, those genes needed to be removed. Because there was no sex change in part of male flowers treated with 5-azaC (M_5-azaC_), DEGs between M_5-azaC_ and M_ck_ could be regarded to be unrelated to sexual differentiation. After removing DEGs between M_5-azaC_ and M_ck_ from DEGs between F_i_ and M_ck_, the remaining genes were directly related to female flower development. The key genes related to female flower development were explored in combination with gene functions.

### 2.6. Quantitative Real-Time PCR (qRT-PCR) Analysis

Thirty-six genes were selected to validate their expression patterns. qPCR primers were designed by prime primer 5 (Appendix A) [219]. The total RNA of each sample was reverse-transcripted into cDNA using PrimeScript™ RT reagent Kit with gDNA Eraser (Takara, Dalian, China). Real-time qPCR was performed using TB Green^®^ Premix Ex Taq™ (Tli RNaseH Plus; Takara, Dalian, China) and analyzed on a BIORAD CFX Real-Time System. Ubiquitin was used for normalization, and the expression ratio was calculated using the 2^−ΔΔCt^ formula.

## 3. Results

### 3.1. Generation of Induced Female Flowers

After treatment with 5-azaC, some male flower buds were induced to transform into female flowers with complete structure and function, while the remaining male flowers had no change in structure and function, and there were no other variation types. Induced female flowers were generated in about 70% of male individuals treated, and induced female flower buds were generated in 7–50% of inflorescence treated among the male individuals in which induced female flowers were generated. Induced female flowers were completely identified as natural female flowers in morphology, structure, and function, characterized in that stamens degenerated to different degrees and ovaries could develop into fruits with seeds (Figure 1).

### 3.2. Transcriptome Characteristics of P. amurense Flower Organs

#### 3.2.1. Results of Transcriptome Assembly and Annotation

The 15 cDNA libraries constructed from all samples were sequenced on an Illumina high-throughput sequencing platform. After filtering the adaptors and low-quality sequences, 335,043,541 high-quality clean reads were generated from the 15 cDNA libraries. The resulting assembly consisted of 263,986 transcripts with an N50 value of 2180 bp. Removing redundant sequences resulted in a total of 86,610 unigenes with an N50 value of 1790 bp. The length distribution of unigenes ranged from 301 to 16,091 bp, nearly 33% of which were 300–500 bp long, approximately 50% of which were also distributed between 500 and 2000 bp, while the remaining 17% were clustered under a size distribution of ≥2000 bp.

A total of 52,231 unigenes (60.30%) were annotated in at least one of the Nr, GO, and KEGG databases (Figure 2). There were 47,588 (54.94%), 31,006 (35.79%), and 17,184 (19.84%) unigenes annotated in the Nr, GO, and KEGG databases, respectively.

According to the species distribution analysis of the Nr database, the top hit was from *Citrus sinensis* (25.4%). Unigenes were also matched significantly with *Citrus clementina* (20.8%) and *Citrus unshiu* (20.4%); additionally, a small number of unigenes matched with *Vitis vinifera* (3.0%) and *Quercus variabilis* (1.6%). The result of Nr annotation suggested *P. amurense* was more similar to *C. sinensis*, *Citrus clementina*, and *Citrus unshiu*, which was consistent with the fact that those four species all belong to Rutaceae (Figure 2).

The GO analysis categorized 31,006 unigenes into three main categories (biological processes, BP; cellular components, CC; and molecular functions, MF). Among the biological processes category, a total of 22,827 unigenes were categorized into 26 functional groups, and “cellular processes” (GO: 0009987) and “metabolic processes” (GO: 0008152) were the predominant groups. Within the cellular components category, a total of 25,399 unigenes were categorized into five functional groups, and cellular anatomical entry (GO: 0110165), intracellular (GO: 0005622), and protein-containing complex (GO: 0032991) were the most overrepresented groups. For the molecular function category, a total of 14,535 unigenes were categorized into 12 functional groups, and a large proportion of unigenes were clustered into binding (GO: 0005488) and catalytic activity (GO: 0003824) (Figure 2).

As the annotation of the KEGG pathway database has suggested, a total of 17,184 unigenes were assigned to 5 main categories, including Cellular Process (6655), Environmental Information Processing (5223), Genetic Information Processing (11,884), Metabolism (27,304), and Organismal System (9580) (Figure 2).

#### 3.2.2. Differential Expression Analysis of Transcriptome between Female and Male Flowers

Differential expression analysis was conducted between female and male flowers (F vs. M) to acquire information on DEGs between the two samples. A total of 16,600 DEGs (19.16%) were identified between F and M, including 7107 up-regulated genes and 9493 down-regulated genes in F compared with M.

The KEGG pathways in which DEGs enriched significantly (corrected *p*-value < 0.05) and the top 30 GO terms distribution for DEGs are shown in Figure 3.

#### 3.2.3. Identification of Sex-Related Genes in Female and Male Flowers and Differential Expression Analysis

To identify genes potentially involved in flower development and sex differentiation, the homologous sequences of sex-related genes from *P. amurense* were acquired using BLASTN (Appendix A). A total of 604 and 584 homologous sequences were acquired from F and M, respectively. F and M shared similar distributions of several genes in the four gene types, with the largest proportion of Group B and the smallest proportion of Group A (Table 1).

To further explore the genes related to flower development and sex differentiation in *P. amurense*, differential expression analysis of sex-related genes between F and M was carried out. Using an FDR of 0.05, a total of 332 sex-related DEGs were identified between F and M, including 172 up-regulated genes and 160 down-regulated genes in F compared with M. The log2FC values of the DEGs ranged from 9.23 to −19.12 (Table 2 and Appendix A).

Some DEGs with female or male-specific expressions might strongly contribute to sexual differentiation. The female-specific genes included early development genes of female organs *ESR2* and *SUP*, chemical metabolism genes *MYBC* and *ATB51*, indoleacetic acid (IAA)-related genes *IAA27* and *SAU50*, ethylene (ETH)-related gene *AIL5*, and gibberellin (GA)-related gene *G3OX3*. The male-specific genes included stamen development gene *GNL2*, pollen maturation and pollen tube growth gene *AGL104*, pollen early development gene *ZAT3*, male gametophyte development gene *MMD1*, pectinesterase-related gene *PEI*, and ATPase-related gene *V-type proton ATPase subunit G1*.

Among A-group DEGs, the sex-determining gene *TOZ19* and genes involved in plant sex regulation, such as *RPP0W*, *PAL3*, *MCM6*, *MCM2*, and *ABA2*, had higher expression in F, while male-determining gene *GPAT3* and the sex regulatory gene *GAST1* had higher expression in M.

Within B-group DEGs, F strongly expressed a few genes responsible for stigma/ovule and other female organ primordia compared with M. These genes included *PIN1*, *PHABULOSA* (*PHB*), *AGL11*, *AINTEGUMENTA* (*ANT*), *AINTEGUMENTA-LIKE6* (*AIL6*), *SUP*, *SHI-RELATED SEQUENCE 5* (*SRS5*), and *SEUSS*. Furthermore, many genes related to female organ subsequent development were up-regulated in F compared with M, including stigma differentiation genes *HEC3* and *SPATULA*; carpel development genes *YABBY2*, *YABBY1*, and *WOX9*; female gamete development genes *PROLIFERA* and *ATH1*; and other pistil development genes *SHOOT MERISTEMLESS*, *FIL*, *RADIALIS-like 1* (*RL1*), and *RADIALIS-like 2* (*RL2*). Additionally, flowering time genes *CRY1* and *COL2* were higher in F. Meanwhile, some of the highly expressed genes in M associated with tapetum/pollen and other male organ early development genes included *BAM2*, *FLOWERING LOCUS D* (*FD*), *ZAT3*, and *HMG-CoA*. In addition, anther/pollen-related genes *MYBS3*, *APG*, *MAC1*, *HXK5*, *LIM2*, *AGL104*, *GAUTE*, *CALS5*, and other male subsequent development genes *GNL2* showed higher expression in M.

For C-group DEGs, methylase-related gene *MET1*; RdDM (RNA silencing and RNA-directed DNA methylation) pathway-related gene *AGO16*; the gene encoding sodium pyruvate cotransporter *BASS2*; sugar and lipid metabolic genes *G3P*, *KPYC1*, and *TPS*; and flavonoid, xylogen, and other chemical metabolic genes *FL3H*, *F3PH*, and *MYBC* were up-regulated in F. Meanwhile, the gene encoding a zinc finger protein *C3H18*; histone lysine methylase-related gene *ATXR6*; sugar metabolic genes *PEI*, *PEL*, *GGAP2*, and *PLDA1*; flavonoid, polyphenol, and other chemical synthesis genes *CCOAMT*, *LAR*, and *MYB111*; and cell cycle pathway-related genes *CDKF4* and *IBS1* were higher in M.

As for D-group DEGs, up-regulated genes in F included IAA-related genes (*ARFs*, *IAAs*, *TIR1*, *LAX3*, and *SAU50*), GA-related genes (*GID1C*, *RGA1*, *SPL16*, *PAT1*, and *G3OX3*), cytokinin (CTK)-related genes (*GI*, *ARR2*, *AHK3*, *ZHD9*, *GTE4*, and *ARR12*), abscisic acid (ABA)-related genes (*ABF2* and *ALDO3*), jasmonic acid (JA)-related genes (*FAD8* and *AOS*), ETH-related genes (*ETR1* and *AIL5*), and a brassinosteroid (BR)-related gene (*BRI1*). Genes with higher expression in M included CTK-related genes (*ARR9* and *CKX7*), ABA-related genes (*ABA 8 hydroxylase 1*, *BEN1*, *SAP11*, *ABAH1*, *CAR4*, *CYP707A2*, and *PYL3*), ETH-related genes (*ACO*, *ERF110*, *ERF109*, and *EREBP 9*), and a JA-related gene (*LOX4*).

### 3.3. Identification of Key Genes Related to Female Flower Development

Some male flowers could be induced into female flowers with no other variation types after treatment with 5-azaC. Key genes related to female flower development in *P. amurense* might be involved in the sex conversion of male flowers. A total of 612, 611, and 606 homologous sequences of sex-related genes were acquired from F_i_, M_ck,_ and M_5-azac_, respectively, using BLASTN. The three samples shared similar distributions in the number of genes in the four gene types (Table 1).

Using an FDR of 0.05, 296 sex-related DEGs were identified between F_i_ and M_ck_, including 142 up-regulated genes and 154 down-related genes in F_i_ compared with M_ck_. The log2FC values of the DEGs ranged from 10.36 to −13.55 (Table 2 and Appendix A).

F_i_ was close to F at the transcriptome level; they shared 216 DEGs, which were regarded as candidate genes for female flower development compared with male flowers. They were more reliable genes for female flower development.

As a demethylation reagent, 5-azaC could provoke extensive physiological changes not limited to sex. Correlation analysis using the Spearman method and principal component analysis showed a good correlation between the replicate sets of F_i_ and M_ck_, but there was an obvious difference among the three biological repeats of M_5-azaC_, which were close to F_i_ or M_ck_, respectively. That suggested that changes in varying degrees had happened within M_5-azaC_ after the treatment with 5-azaC, but these changes might involve other biological activities unrelated to sex formation (Figure 4). Therefore, the DEGs between M_5-azaC_ and M_ck_ were less likely to be key genes and could be regarded as not linked with sex formation. A total of 28 such genes were removed from 216 candidate genes of female flowers, and finally, 188 candidate genes were screened out (Figure 5 and Appendix A). The candidate genes were mainly involved in sex regulation, the initiation and development of the floral organ, biochemical metabolic pathways, and plant hormones. Combining with their functions, genes associated with sex regulation and floral initiation, including *RPP0W*, *PAL3*, *MCM2*, *MCM6*, *SUP*, *ANT*, *AIL6*, *AGL11*, *SEUSS*, *SRS5*, and *ESR2*, were preliminarily considered to be the key genes for female flower development.

### 3.4. Verification of DEGs Using qRT-PCR Analysis

We paid attention to the following two types of genes when selecting genes to be validated. One was A-group genes because the genes involved in sex regulation might contribute strongly to sex determination. Another was up-regulated genes in F_i_, considering that demethylation is more likely to improve the expression levels of target genes. Therefore, 36 genes, including A-group genes and up-regulated genes in F_i_, were selected for qRT-PCR validation (Figure 6).

## 4. Discussion

RNA-Seq has been a common technique applied in plants for exploring DEGs between female and male flowers, such as papaya [220] and *Cannabis sativa* [68]. However, there have been few reports on genes related to female flower development in which demethylation was used to generate induced female flowers and further analysis was performed combining sex-related genes known in other plants. In this study, we succeeded at inducing male flowers into female flowers using the demethylation reagent 5-azaC. After acquiring sex-converted materials, 188 candidate genes of female flower development were identified with further analysis.

In *P. amurense*, the female flowers are present as bisexual tissue at the initial stage, and then the stamen stops developing, forming a unisexual flower. This process requires many specific genes to participate in each development stage. The 188 candidate genes acquired in this study are involved in sex regulation, flower development, biochemical pathway, and hormone regulation. The up-regulated genes are more likely to be influenced by demethylation. Among them, *PAL3*, *RPP0W*, *MCM2*, *MCM6*, and *SRS5* might be involved in sexual differentiation regulation in *P*. *amurense* during the sex-determining stage. *PAL3*, encoding a key player in the phenylpropanoid pathway, was detected as a candidate gene for female flower development in *Populus tomentosa* with a sex-specific methylation alteration [221]. *RPP0W*, identified in the SDR (sex-determining region) cassette occurring in nearly all females and never in males, potentially regulated the sexual differentiation in *Fragaria* [162]. *MCM2* and *MCM6*, two genes involved in the cell cycle pathway, had much higher expression in the gynoecious line than those in weak female cucumbers and were considered key candidate genes of sexual differentiation in cucumbers [166]. *SRS5* had higher expression in early gynoecious inflorescence buds than monoecious plants in *Jatropha curcas*, which was considered a candidate regulator of the sex determination [222].

During the flowering stage in *P. amurense*, *ANT*, *AIL6*, *SEUSS*, *AGL11*, *SUP*, *PIN1*, and *ESR2* might promote the initiation of female floral organ primordia. *ANT* and its paralog *AIL6* are two key regulators of flower development, including floral organ initiation, identity specification, growth, and patterning. On the one hand, they are important for establishing the flower primordia, such as ovule and female gametophyte in *Arabidopsis*. On the other hand, they participate in subsequent female flower development by regulating target genes including *SPATULA*, *YABBY*, *SEP3*, *PHB*, *AS1*, *REV*, and *FIL* and genes associated with hormones identified in *P. amurense* [223,224,225]. *SEUSS* cooperates with *ANT* in a partially redundant manner to regulate the expression of downstream genes critical for the formation of ovules [226]. *AGL11* participates in the early development of ovules, which was demonstrated in many species, including *Arabidopsis thaliana* and *Punica granatum* [227,228]. *SUP* and *PIN1* were demonstrated to be involved with reproductive phase transition and female flower transition by promoting the abortion of male flower primordia in *Jatropha curcas* [229]. Furthermore, *SUP* plays a role in maintaining the boundaries between stamens and carpels and regulating the development of outer ovule integument in *Arabidopsis* [230]. *ESR2* is a member of transcription factor BOL/DRNL/ESR2/SOB, which is expressed at the very early stages in aerial organ formation and has been proposed to be a marker for organ founder cells [231]. The results of this study suggest that the genes above were likely to participate in the early establishment of female flowers, which have been regarded as key genes for female flower development and likely make significant contributions to female formation in *P. amurense*. It is worth mentioning that *PAL3*, *MCM2*, *MCM6*, *AGL11*, *ANT*, *AIL6*, *SRS5*, *ESR2*, and *SUP* have shown differential expression during our research aimed at early flowers (the result is unpublished), which could make the results of this study more convincing to some degree. Moreover, *CYC2CL*, *RL1*, *RL2*, *PROLIFERA*, *FBP2*, *YABBY1*, *YABBY2*, *FIL*, *AS1,* and *PHB* might be involved in subsequent female flower development in *P. amurense*. Two *Cyc2CL* transcripts (*Cyc2CL-1* and *Cyc2CL-2*) regulate petal development and stamen abortion and are important for the ray floret (without stamens) development in the chrysanthemum [211]. *RL*-like gene, also named *RAD*, was one of the shared genes in both pathways leading to reversion from male to hermaphrodite flowers in hexaploid *D. kaki* and ectopic overexpression of *DkRAD* in model plants resulting in hypergrowth of the gynoecium [232]. *PROLIFERA* is necessary for megagametophyte and embryo development [233]. *FBP2*, a MADS-box gene involved in flower development, represents the same E function as *SEP3* in *Arabidopsis* [234]. It has been demonstrated that *YABBY* genes participate in ovule development. Among *YABBY* genes, *YABBY2* plays a critical role in ovule development and is expressed in different ovule development stages, while *YABBY1* has the highest expression in the Megaspore Mother Cell stage [101]. It was reported that *FIL*, *AS1*, *PHB*, and *YABBY1* interact with *ANT* and *SEUSS* to regulate ovule development and promote organ polarity. For example, *ANT* combined with the *YABBY* gene *FIL* to promote organ polarity by up-regulating the expression of the adaxial-specifying HD-ZIP gene *PHABULOSA*. The *SEUSS*/*LUG* coregulator complex physically interacts with *YABBY1* and/or *ANT* to regulate adaxial identity genes *PHB* and *REV* to promote ovule and carpel growth [223,224,225,226,235]. Therefore, the genes above might form a flower development network, with *ANT* and *SEUSS* as initiation factors in *P. amurense*. Herein, a simple model of female flower development in *P. amurense* is proposed as a reference for future research (Figure 7).

Additionally, *FMO3*, *ATPD*, *WRKY21*, and *exostosin* are located in the sex-determined region of plants with unknown specific functions [148,176,214]. Their specific functions need further study. Male-related genes *TOZ19* and *SERK1* were up-regulated in female and induced female flowers compared to male flowers [45,207]. The genes have not been well studied and whether they have the same role in *P. amurense* needs further verification. Furthermore, some male-related genes, including *GPAT3*, *ZAT3,* and *FD* [30,83,89], were down-regulated, which might promote female development by suppressing male development.

Biochemical pathway-related genes and hormone-related genes might participate in each stage of female flower development in *P. amurense*. The two types of genes might contribute to extensive physiological activities. Their specific roles in *P. amurense* have not yet been well illustrated in this study.

Fan et al. discovered monoecious *P. amurense* for the first time and preliminarily explored the sexual formation and differentiation mechanism [30]. The discovery of monoecious *P. amurense* led to the sexual differentiation mechanism of *P. amurense* at the epigenetic level. This suggested that we should carefully observe the individual characteristics of plants, from which we can obtain inspiration. In this study, a precise method of demethylation was applied to further demonstrate that epigenetics was indeed involved in the sex regulation of *P. amurense*, and DNA methylation was one of its regulation modes. Sex transition from male to female flowers with complete structure and function through demethylation suggested that demethylation might influence key genes for the sex differentiation in *P. amurense*. Among the genes analyzed, *PAL3* and *HUA1* were reported to regulate plant sex by DNA methylation [8,221]. Indeed, the performance of *PAL3* and *HUA1* in *P. amurense* after demethylation was consistent with the function reported. Therefore, these two genes deserve attention in subsequent studies.

Male-related genes (including *GPAT3*, *ASHR3*, *ARAD1*, *GDSL*, *MCM2*, *LAX2*, *AP2*, *OFP*, and *MYB36*) and female-related genes (including *AUX*, *AGL12*, *60s ribosomal protein*, *BRI1*, *CKX7*, *AGL8*, *TCP1*, *CCR4-NOT*, *NDH*, *PGDH*, and *NDUFA12*) were selected during the previous research of bark in young *P. amurense* [30]. Among them, male-related genes (including *GPAT3*, *ASHR3*, *ARAD1*, and *GDSL*) and the female-related gene *BRI1* were consistent with the results in this study, suggesting that the genes above might play a role in early sex differentiation from the young stage of *P. amurense*.

## 5. Conclusions

Male flowers of *P. amurense* could be induced to convert into female flowers with complete structure and function with 5-azaC treatment, demonstrating that epigenetics is involved in the sex regulation of *P. amurense*. Potential sex-related genes were acquired from the transcriptome of *P. amurense* by referring to sex-related genes reported in the literature. Sex-related DEGs between female flowers (including F and F_i_) and male flowers were identified by comparing female and male flowers. Further, 188 candidate genes of female flower development were screened out, including sex-regulatory genes, genes related to the formation and development of sexual organs, genes related to biochemical pathways, and hormone-related genes. Combined with gene functions and qRT-PCR analysis, *RPP0W*, *PAL3*, *MCM2*, *MCM6*, *SUP*, *ANT*, *AIL6*, *AGL11*, *SEUSS*, *SRS5*, and *ESR2* were preliminarily considered to be the key genes for female flower development.

## Figures and Tables

**Figure 1 genes-14-00661-f001:**
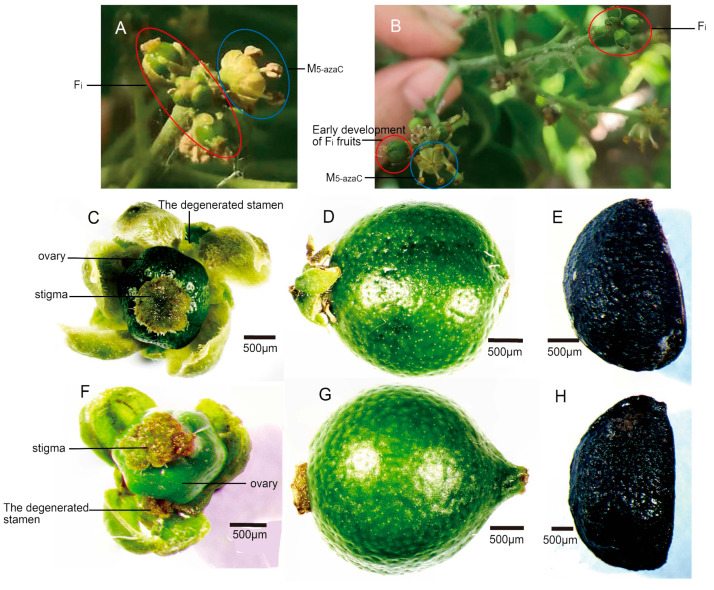
Induced female flowers with complete structure and function could develop into fruits with seeds. (**A**,**B**) Sex conversion of some male flowers after treatment with 5-azaC. (**C**,**F**) F and F_i_, respectively. (**D**,**G**) Fruits developed from F and F_i_, respectively. (**E**,**H**) Seeds in fruits developed from F and F_i_, respectively.

**Figure 2 genes-14-00661-f002:**
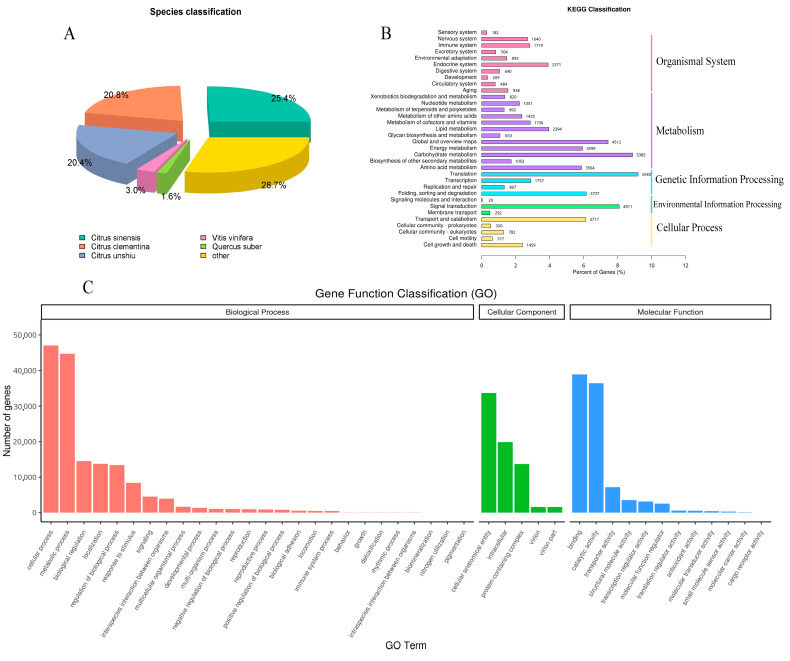
The result of annotation. (**A**) Species distribution analysis of the Nr database. (**B**) The annotation of the KEGG pathway database. (**C**) The annotation of the GO database.

**Figure 3 genes-14-00661-f003:**
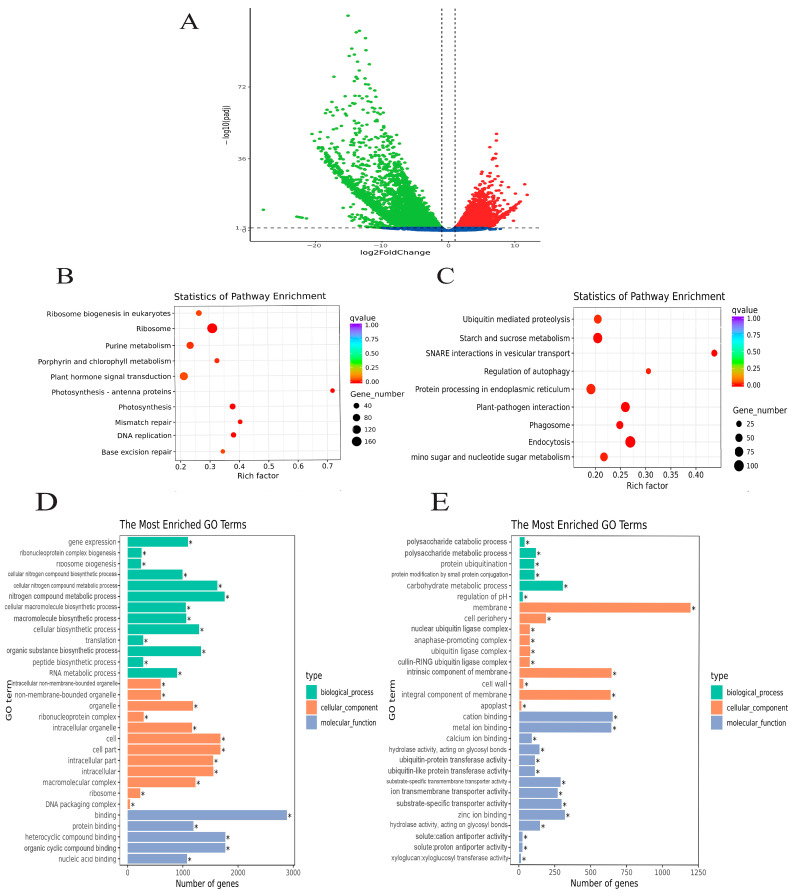
Results of differential expression analysis between F and M. (**A**) The volcanic map of DEGs of transcriptome between F and M. Red and green dots present female highly expressed and male highly expressed genes, respectively, and blue dots mean no significant difference. (**B**,**C**) Enrichment of female highly expressed and male highly expressed genes in KEGG pathways. (**D**,**E**) Enrichment of female highly expressed and male highly expressed genes in the GO database. * significant enrichment (corrected *p*-value < 0.05).

**Figure 4 genes-14-00661-f004:**
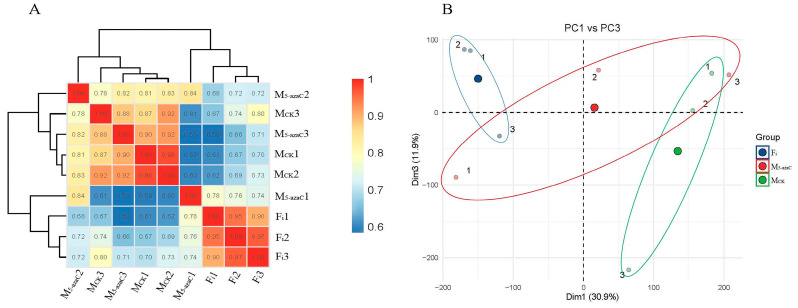
Correlation analyses showing the relationship between samples and replicates. (**A**) Correlation matrix. (**B**) Principal component analysis.

**Figure 5 genes-14-00661-f005:**
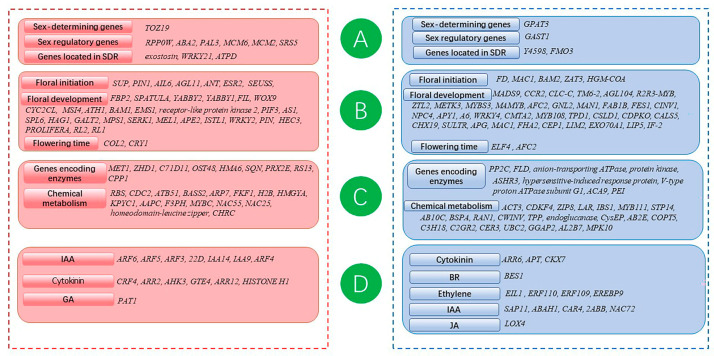
Candidate genes of female flower development (red and blue represent up-regulated and down-regulated genes in female flowers, respectively).

**Figure 6 genes-14-00661-f006:**
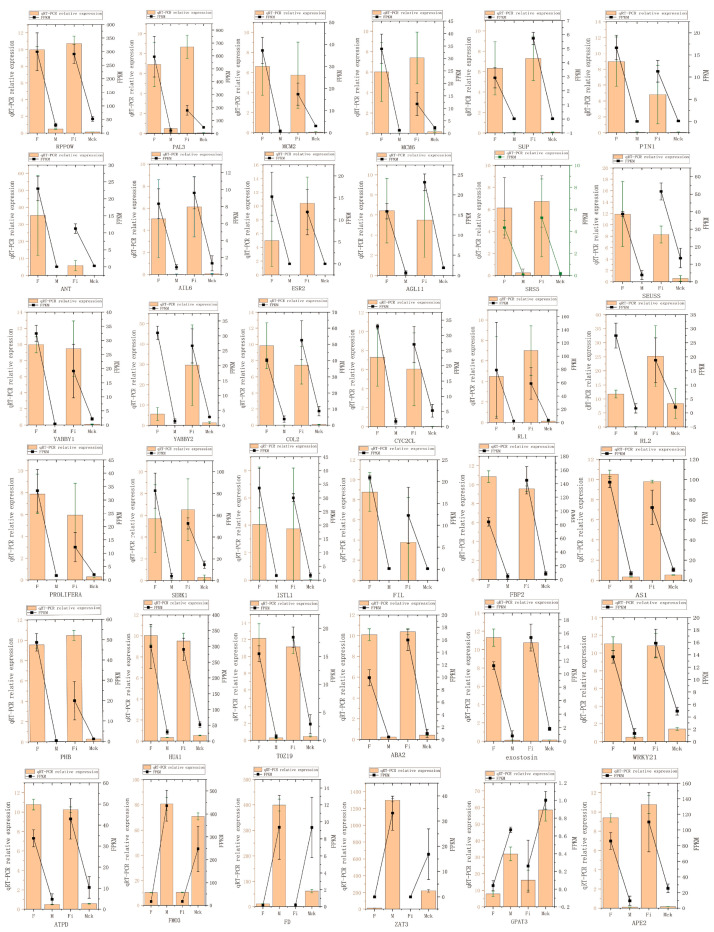
qRT-PCR validation of DEGs. Some A-group genes (including *RPP0W*, *PAL3*, *MCM2*, *MCM6*, *TOZ19*, *ABA2*, *exostosin*, *WRKY21*, and *ATPD*) and some B-group genes (including *SUP*, *PIN1*, *ANT*, *AIL6*, *ESR2*, *AGL11*, *SRS5*, *SEUSS*, *YABBY1*, *YABBY2*, *COL2*, *CYC2CL*, *RL1*, *RL2*, *PROLIFERA*, *SERK1*, *ISTL1*, *FIL*, *FBP2*, *AS1*, *PHB*, *HUA1*, and *APE2*) were up-regulated in female flowers. A-group genes (including *FMO3* and *GAPT3*) and B-group genes (including *FD* and *ZAT3*) were down-regulated in female flowers.

**Figure 7 genes-14-00661-f007:**
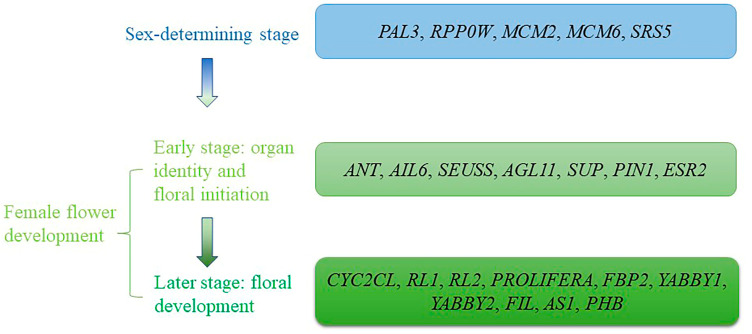
A simple model of female flower development shows the potential genes involved in each stage.

**Table 1 genes-14-00661-t001:** The number of sex-related genes acquired from all samples.

	F	M	F_i_	M_5-azaC_	M_ck_
Sex regulatory genes (Group A)	44	43	45	45	44
Genes related to floral initiation and development (Group B)	250	239	255	251	248
Genes related to biochemical metabolic pathways (Group C)	177	177	180	180	183
Plant hormones-related genes (Group D)	133	127	132	135	131
Total	604	586	612	611	606

**Table 2 genes-14-00661-t002:** The number of DEGs in F vs. M and F_i_ vs. M_ck_ comparisons.

Expression	log2 FC	FDR	Number of DEGs	
			F vs. M	F_i_ vs. M_ck_
Up-regulated	≥ 1	≤0.05 ≤0.05	166	129
	1 > log2 FC > 0		6	13
Down-regulated	0 > log2 FC > −1	≤0.05	3	16
	≤−1	≤0.05	157	138
Total			332	296

## Data Availability

All raw reads of these libraries have been deposited in the National Center for Biotechnology Information (NCBI) Sequence Read Archive (SRA) database under the BioProject accession PRJNA934260 (https://www.ncbi.nlm.nih.gov/sra/PRJNA934260) (accessed on 15 February 2023).

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
