# Peer review of "Identifying Genes Associated with Female Flower Development of Phellodendron amurense Rupr. Using a Transcriptomics Approach"

_genes, 2023, doi:10.3390/genes14030661_

Round 1
Reviewer 1 Report
Dear Editor,
Thank you for considering me as a reviewer for your valued journal. I appreciate the work involved in this manuscript. The work described in the manuscript is original and scientifically valid. It is well organized with clear figures and illustrations. The authors compared the transcriptome data of female flowers and 5-azaC-induced female flowers of Phellodendron amurense Rupr. with the male ones, together with previously reported sex-related genes of different plant species. The consequence of this study suggests that the sex P. amurense is regulated by epigenetics and provides important information for the sexual differentiation mechanism of P. amurense. Therefore, the manuscript contains some interesting conclusions and is found to be accepted with minor revision. Some suggestions for the authors are noted.
Best regards
Author Response
Response to Reviewer 1 Comments
Dear reviewer,
We would like to thank you for the valuable comments on our manuscript. These suggestions and opinions have played an important role in improving our manuscript. We have revised the article after careful consideration of the comments and suggestions, and we look forward to meeting your expectations.
Primerily, sincere gratitude would be expressed to you for recognition of our work. Here our responses to each comment are as follows.
Point 1: The cited references must be improved.
Response 1: Thank you for the suggestion. As is stated in the instructions for authors, citations and references in supplementary files are permitted provided that they also appear in the main text and in the reference list. Because citations were needed in Tables S1 (Sex-related genes reported) and S6 (The list of candidate genes with specific functions), it was necessary to list references in main text and reference list according to the instructions for authors. We will communicate with the editor later. If it should be allowed not to appear in the main text and in the reference list, we would transfer part of references to supplementary files.
Yours,
Zhao Zhang

Reviewer 2 Report
This manuscript entitled Identifying genes associated with female flower development of Phellodendron amurense Rupr. using a transcriptomics approach by Lihong He, Yongfang Fan, Zhao Zhang, Xueping Wei and Jing Yu is a nice piece of work. Manuscript provides some interesting findings regarding sex regulation and flower development of P. amurense. However, to improve the overall quality of this work I suggest the following: It is better to write down how 5-azaC was applied to flower buds (injected, sprayed, what was the volume of 1 mM 5-azaC, how many times applied during dormancy and growth period?). I noticed that lines 23, 204, 207 and 446 need to be revised to correct grammar, writing and minor mistakes. I also noted that there are several sentences repeated in the manuscript. For instance, lines 264-267 were basically very similar/same to lines 308-312. There are some other similar or same sentences repeated throughout the manuscript. I suggest many references listed from 27 to 215, along with 4 and 8 should be removed since there is no information given from these studies. One more suggestion keywords are composed of words given in the title.
Author Response
Response to Reviewer 2 Comments
Dear reviewer,
We would like to thank you for the valuable comments on our manuscript. These suggestions and opinions have played an important role in improving our manuscript. We have revised the article after careful consideration of the comments and suggestions , and we look forward to meeting your expectations.
Primerily, sincere gratitude would be expressed to you for recognition of our work. Here our responses to each comment are as follows.
Point 1: It is better to write down how 5-azaC was applied to flower buds (injected, sprayed, what was the volume of 1 mM 5-azaC, how many times applied during dormancy and growth period?).
Response 1: The details of 5-azaC treatment were added as suggested. Thank you!
Point 2: I noticed that lines 23, 204, 207 and 446 need to be revised to correct grammar, writing and minor mistakes.
Response 2: Thank you for pointing out the minor mistakes. We have rectified that.
Point 3: I also noted that there are several sentences repeated in the manuscript. For instance, lines 264-267 were basically very similar/same to lines 308-312.
Response 3: We have appropriately deleted sentence repeats in lines 308-314, 360-361, and 364-368 according to your suggestions.
Point 4: I suggest many references listed from 27 to 215, along with 4 and 8 should be removed since there is no information given from these studies.
Response 4: Thank you for the suggestion. As is stated in the instructions for authors, citations and references in supplementary files are permitted provided that they also appear in the main text and in the reference list. Because citations were needed in Tables S1 (Sex-related genes reported) and S6 (The list of candidate genes with specific functions), it was necessary to list references in main text and reference list according to the instructions for authors. We will communicate with the editor later. If it should be allowed not to appear in the main text and in the reference list, we would transfer part of references to supplementary files.
Point 5: One more suggestion keywords are composed of words given in the title.
Response 5: Considering your suggestions, we have modified keywords as “DNA methylation; Phellodendron amurense; female flower development; transcriptome; epigenetics; plant sex differentiation; sex-related genes”.
Yours,
Zhao Zhang
